# Unifying Graph Convolutional Networks as Matrix Factorization

## Abstract

In recent years, substantial progress has been made on graph convolutional networks (GCN). In this paper, for the first time, we theoretically analyze the connections between GCN and matrix factorization (MF), and unify GCN as matrix factorization with co-training and unitization. Moreover, under the guidance of this theoretical analysis, we propose an alternative model to GCN named **Co**-training and **U**nitized **M**atrix **F**actorization (CUMF). The correctness of our analysis is verified by thorough experiments. The experimental results show that CUMF achieves similar or superior performances compared to GCN. In addition, CUMF inherits the benefits of MF-based methods to naturally support constructing mini-batches, and is more friendly to distributed computing comparing with GCN. The distributed CUMF on semi-supervised node classification significantly outperforms distributed GCN methods. Thus, CUMF greatly benefits large scale and complex real-world applications.

## 1 Introduction

In recent years, works on graph convolutional networks (GCN) (Kipf & Welling, 2017) have achieved great success in many graph-based tasks, e.g., semi-supervised node classification (Kipf & Welling, 2017), link prediction (Zhang & Chen, 2018) and recommendation systems (Ying et al., 2018). GCN defines a graph convolution operation, which generates the embedding of each node by aggregating the representations of its neighbors. Given a graph, GCN performs the graph convolution operation layer by layer to obtain the final node representations, which will be passed to neural networks to support various tasks. To perform GCN on large scale graphs in constrained memory or distributed computing environments, different sampling methods have been proposed, such as neighbor sampling (Hamilton et al., 2017) and importance sampling (Chen et al., 2018b). Instead of sampling, Cluster-GCN (Chiang et al., 2019) proposes an approach to convert computation on a huge matrix to computing on a set of small matrices. However, these methods still suffer from performance loss when conducting distributed computing. To take use of various contextual information on edges in a graph, Relational GCN (RGCN) (Schlichtkrull et al., 2018) extends neighbor aggregation by using edge types in link prediction. Besides the edge types, Edge-enhanced Graph Neural Networks (EGNNs) (Gong & Cheng, 2019) takes more contextual features into consideration. However, in general, GCN still has the efficiency problem when facing complex forms of contextual information.

Besides GCN, graph embedding methods (Perozzi et al., 2014; Tang et al., 2015b;a; Grover & Leskovec, 2016) are also widely applied. In general, these methods rely on first-order and second-order proximity to embed very large information networks into low-dimensional vector spaces. The first-order proximity in a graph is the local pairwise proximity between two vertices, and the second-order proximity between a pair of vertices in a graph is the similarity between their neighborhood structures. As for GCN, previous work shows that the graph convolution operation is actually a special form of Laplacian smoothing (Li et al., 2018). Thus, as the converging of the model, the smoothing process can keep the final representation of a node more and more similar to those of its neighbors. Therefore, GCN is consistent with graph embedding methods in capturing the structural information. According to previous work (Qiu et al., 2018), graph embedding methods have been successfully unified as matrix factorization (MF). Thus, we believe that there might be some connections between GCN and MF. Meanwhile, comparing with GCN, MF-based methods are extremely flexible and suitable for distributed computing (Gemulla et al., 2011; Zhuang et al., 2013; Yu et al.,

2014). MF-based methods are also easy and efficient to be extended to tasks with complex forms of contextual information on graph edges (Rendle et al., 2011; Rendle, 2012; Jamali & Lakshmanan, 2013; Shi et al., 2014; Liu et al., 2015). Thus, if we can unify the GCN model as a special form of MF, large scale and complex real-world applications will benefit from this.

In this paper, we theoretically reveal the connections between GCN and MF, and unify GCN as matrix factorization with co-training and unitization in section 2. Here, the co-training process means co-training with the classification task of labeled nodes as in (Weston et al., 2012; Yang et al., 2016), and the unitization indicates conducting vector unitization on node representations. Then, under the guidance of our theoretical analysis, we formally propose an alternative model to GCN named **C**o-training and **U**nitized **M**atrix **F**actorization (CUMF) in section 3. Extensive experiments are conducted on several real-world graphs, and show co-training and unitization are two essential components of CUMF. Under centralized computing settings, CUMF achieves similar or superior performances comparing with GCN. These observations strongly verify the correctness of our theoretical analysis. Moreover, GCN performs poor on dense graphs, while CUMF has great performances. This is may caused by the over-smoothing of graph convolution on dense graphs, while CUMF can balance the smoothing of neighbours and the classification of labeled nodes through the co-training process. Experiments under distributed computing settings are also conducted, and distributed CUMF significantly outperforms the state-of-the-art distributed GCN method, i.e., cluster-GCN (Chiang et al., 2019). Thus, CUMF is extremely friendly to large scale real-world graphs. Meanwhile, lots of works have been done to model contextual information in MF-based methods (Rendle et al., 2011; Rendle, 2012; Jamali & Lakshmanan, 2013; Shi et al., 2014; Liu et al., 2015), which have shown great effectiveness, efficiency and flexibility.

## 2 GCN AS CO-TRAINING AND UNITIZED MF

In this section, we plan to theoretically unify GCN as a specific form of matrix factorization. First, we need to start from the analysis of how node representations are learned in GCN. According to the definition in previous work (Kipf & Welling, 2017), we can formulate each layer of GCN as

$$\mathbf{H}^{(l+1)} = \sigma \left( \widetilde{\mathbf{D}}^{-\frac{1}{2}} \widetilde{\mathbf{A}} \widetilde{\mathbf{D}}^{-\frac{1}{2}} \mathbf{H}^{(l)} \mathbf{W}^{(l)} \right), \tag{1}$$

where $\widetilde{\mathbf{A}} = \mathbf{A} + \mathbf{I}_N$ is the adjacency matrix of the graph $G$ with added self-connections, $\mathbf{I}_N$ is the identity matrix for $N$ nodes in graph $G$, $\widetilde{\mathbf{D}}$ is a diagonal degree matrix with $\widetilde{\mathbf{D}}_{i,i} = \sum_j \widetilde{\mathbf{A}}_{i,j}$, $\mathbf{H}^{(l)}$ is the representation of each node at layer $l$, $\mathbf{W}^{(l)}$ is a layer-specific trainable weight matrix, and $\sigma(\cdot)$ denotes an activation function (such as $\mathrm{ReLU}(\cdot) = \max(0, \cdot)$).

For a node classification task, we can obtain a classification loss

$$l_{\mathrm{class}} = \mathrm{CrossEntropy}\left( \mathbf{Y}, \mathrm{softmax}\left( \mathbf{H}^{(-1)} \right) \right), \tag{2}$$

where $\mathbf{Y}$ is the ground truth labels for the classification task, $\mathbf{H}^{(-1)}$ is the representation of each node at the final layer of GCN. Via optimizing Eq. (2), the cross-entropy error of the node classification task can be minimized, and the GCN model can be learned.

As in the implementation in previous work (Kipf & Welling, 2017; Veličković et al., 2017; Wu et al., 2019), there is no activation function on the last layer of GCN, and the final representations can be formulated as

$$\mathbf{H}^{(-1)} = \widetilde{\mathbf{D}}^{-\frac{1}{2}} \widetilde{\mathbf{A}} \widetilde{\mathbf{D}}^{-\frac{1}{2}} \mathbf{H}^{(-2)} \mathbf{W}^{(-2)}. \tag{3}$$

In (Li et al., 2018), GCN has been proven to be a special form of Laplacian smoothing. The final representations in Eq. (3) tends to converge to

$$\mathbf{H}^{(-1)} = \widetilde{\mathbf{D}}^{-\frac{1}{2}} \widetilde{\mathbf{A}} \widetilde{\mathbf{D}}^{-\frac{1}{2}} \mathbf{H}^{(-1)}, \tag{4}$$

which is an approximate solution of the final representations in GCN. Specifically, for each node $i$ in graph $G$, the approximate solution of the corresponding final representation is

$$h_i^{(-1)} = \sum_j \frac{1}{\sqrt{(d_i + 1)(d_j + 1)}} \widetilde{\mathbf{A}}_{i,j} h_j^{(-1)}, \tag{5}$$

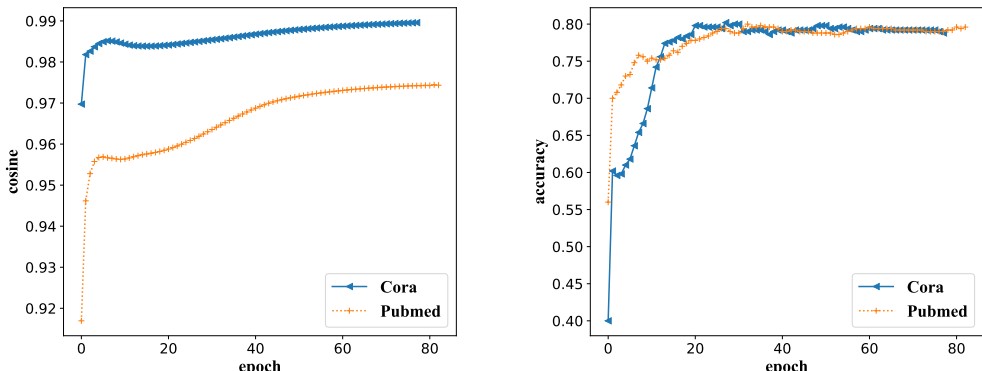

(a) The average cosine similarity between nodes in the graph and their neighbors during the training of GCN.

(b) The accuracy curves during the training of GCN.

Figure 1: Consistency estimation between the cosine similarity and the convergence of GCN during training procedure on the Cora dataset and the Pubmed dataset.

from which we have

$$h_i^{(-1)} = \sum_j \frac{1}{\sqrt{(d_i+1)(d_j+1)}} \mathbf{A}_{i,j} h_j^{(-1)} + \frac{1}{d_i+1} h_i^{(-1)}. \tag{6}$$

Then, the representation of node $i$ can be obtained as

$$h_i^{(-1)} = \sum_j \frac{1}{d_i} \sqrt{\frac{d_i+1}{d_j+1}} \mathbf{A}_{i,j} h_j^{(-1)}, \tag{7}$$

where $d_i$ is the degree of node $i$.

According to above analysis, to train an approximate GCN model with one loss, which simultaneously models the structure of graph convolution and the node classification task, we can minimize the following loss function

$$l = \alpha \, l_{\text{class}} + (1-\alpha) \, l_{structure}, \tag{8}$$

where $\alpha$ a hyper-parameter to control the balance between two losses, and the structure loss $l_{structure}$ refers to

$$l_{structure} = \sum_{i \in I} \text{dis}\left(h_i^{(-1)}, \sum_j \frac{1}{d_i} \sqrt{\frac{d_i+1}{d_j+1}} \mathbf{A}_{i,j} h_j^{(-1)}\right), \tag{9}$$

where $I$ denotes the set of all the nodes in graph $G$, and $\text{dis}(\cdot, \cdot)$ is a distance measurement. Here, we apply the commonly used cosine similarity and obtain

$$l_{structure} = \sum_{i \in I} \cos\left(h_i^{(-1)}, \sum_j \frac{1}{d_i} \sqrt{\frac{d_i+1}{d_j+1}} \mathbf{A}_{i,j} h_j^{(-1)}\right), \tag{10}$$

which is equivalent to

$$l_{structure} = -\sum_{i \in I} \frac{h_i^{(-1)} \left(\sum_j \frac{1}{d_i} \sqrt{\frac{d_i+1}{d_j+1}} \mathbf{A}_{i,j} h_j^{(-1)}\right)^{\top}}{\left\|h_i^{(-1)}\right\| \left\|\sum_j \frac{1}{d_i} \sqrt{\frac{d_i+1}{d_j+1}} \mathbf{A}_{i,j} h_j^{(-1)}\right\|}. \tag{11}$$

To verify whether cosine similarity is consistent with the convergence of GCN during the training procedure, we conduct empirical experiments and train GCN models on the Cora dataset and the Pubmed dataset. Figure 1 demonstrates the average cosine similarity between nodes in the graph

and their neighbors, as well as the convergence curves estimated by accuracy during the training of GCN on the two datasets. It is obvious that, the curves on the same dataset share similar tendency. That is to say, as we train a GCN model, the cosine similarity between nodes in the graph and their neighbors is being optimized. This strongly proves that cosine similarity is consistent with the convergence of the GCN model.

Then, to simplify the form of Eq. (11), we conduct vector unitization in the learned representations $\mathbf{H}^{(-1)}$, and thus each representation $h_i^{(-1)}$ has similar l2-norm. Accordingly, through unitization, Eq. (11) is equivalent to

$$l_{structure} = -\sum_{i \in I} h_i^{(-1)} \left( \sum_j \frac{1}{d_i} \sqrt{\frac{d_i+1}{d_j+1}} \, \mathbf{A}_{i,j} \, h_j^{(-1)} \right)^{\top}, \tag{12}$$

which leads to

$$l_{structure} = -\sum_{i \in I} \sum_{j \in C_i} \frac{1}{d_i} \sqrt{\frac{d_i+1}{d_j+1}} \, \mathbf{A}_{i,j} \, v_i \, v_j^{\top}, \tag{13}$$

where $C_i$ denotes all the nodes that node $i$ is connected to, and $v_i = h_i^{(-1)}$ for simplicity. Moreover, for better optimization, we can incorporate negative log likelihood and minimize the following loss function equivalently to Eq. (13)

$$l_{structure} = -\sum_{i \in I} \sum_{j \in C_i} \frac{1}{d_i} \sqrt{\frac{d_i+1}{d_j+1}} \, \mathbf{A}_{i,j} \log \left( \lambda \left( v_i \, v_j^{\top} \right) \right), \tag{14}$$

where $\lambda(\cdot) = \text{sigmoid}(\cdot)$.

So far, the structure loss $l_{structure}$ is equivalent to factorizing all the positive edges in the graph $G$ weighted by $\beta_{i,j} = d_i^{-1} (d_i+1)^{1/2} (d_j+1)^{-1/2}$.

Usually, in graph embedding methods (Perozzi et al., 2014; Tang et al., 2015b;a; Grover & Leskovec, 2016), negative edges sampling is used, for better convergence. Thus, we can randomly sample negative edges for each edge in graph $G$. Following previous work in unifying word embedding (Levy & Goldberg, 2014) and graph embedding (Qiu et al., 2018) as implicit matrix factorization, we can rewrite Eq. (14) as

$$l_{structure} = -\sum_{i \in I} \sum_{j \in C_i} \left( \beta_{i,j} \, \mathbf{A}_{i,j} \log \left( \lambda \left( v_i \, v_j^{\top} \right) \right) + k \, \mathbb{E}_{j' \sim P_G} \left[ \beta_{i,j'} \log \left( \lambda \left( - v_i \, v_{j'}^{\top} \right) \right) \right] \right), \tag{15}$$

and

$$l_{structure} = -\sum_{i \in I} \sum_{j \in C_i} \beta_{i,j} \, \mathbf{A}_{i,j} \log \left( \lambda \left( v_i \, v_j^{\top} \right) \right) - k \sum_{i \in I} d_i \, \mathbb{E}_{j' \sim P_G} \left[ \beta_{i,j'} \log \left( \lambda \left( - v_i \, v_{j'}^{\top} \right) \right) \right], \tag{16}$$

where $k$ is the number of negative samples for each edge, and $P_G$ denotes the distribution that generates negative samples in graph $G$. For each node $j$, $P_G(j) = d_j / |G|$, where $|G|$ is the number of edges in graph $G$. Then, we can explicitly express the expectation term as

$$\mathbb{E}_{j' \sim P_G} \left[ \beta_{i,j'} \log \left( \lambda \left( - v_i \, v_{j'}^{\top} \right) \right) \right] = \sum_{j' \in I} \frac{\beta_{i,j'} \, d_{j'}}{|G|} \log \left( \lambda \left( - v_i \, v_{j'}^{\top} \right) \right), \tag{17}$$

from which we have

$$\mathbb{E}_{j' \sim P_G} \left[ \beta_{i,j'} \log \left( \lambda \left( - v_i \, v_{j'}^{\top} \right) \right) \right] = \frac{\beta_{i,j} \, d_j}{|G|} \log \left( \lambda \left( - v_i \, v_j^{\top} \right) \right) + \sum_{j' \in I \setminus \{j\}} \frac{\beta_{i,j'} \, d_{j'}}{|G|} \log \left( \lambda \left( - v_i \, v_{j'}^{\top} \right) \right). \tag{18}$$

Via combining Eq. (16) and Eq. (18), we can obtain the local structure loss $l_{structure}$ for a specific edges $(i, j)$

$$l_{structure}(i, j) = - \beta_{i,j} \, \mathbf{A}_{i,j} \log \left( \lambda \left( v_i \, v_j^{\top} \right) \right) - \frac{k \, \beta_{i,j} \, d_i \, d_j}{|G|} \log \left( \lambda \left( - v_i \, v_j^{\top} \right) \right). \tag{19}$$

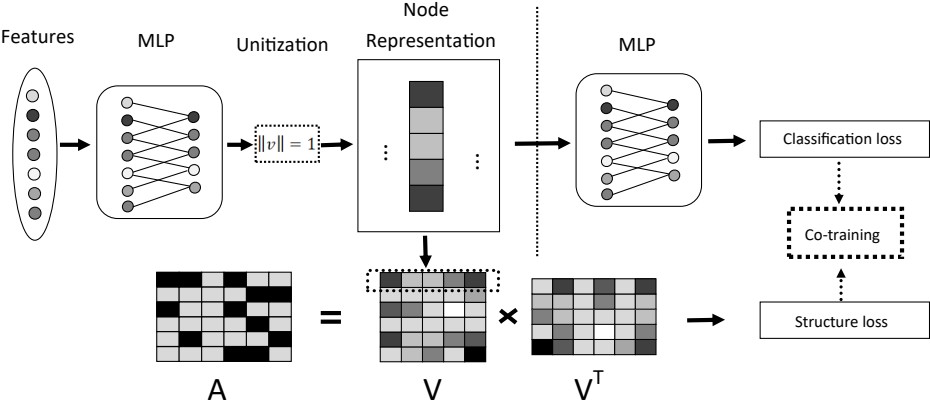

Figure 2: Overview of our proposed CUMF model.

To optimize the objective, we need to calculate the partial derivative of $l_{structure}$ with respect to $v_i\,v_j^T$

$$\frac{\partial\,l_{structure}}{\partial\left(v_i\,v_j^\top\right)} = -\,\beta_{i,j}\,\mathbf{A}_{i,j}\,\lambda\left(-\,v_i\,v_j^\top\right) + \frac{k\,\beta_{i,j}\,d_i\,d_j}{|G|}\lambda\left(v_i\,v_j^\top\right). \tag{20}$$

After setting Eq. (20) to zero, we can obtain

$$e^{2\,v_i\,v_j^\top} - \left(\frac{|G|\,\mathbf{A}_{i,j}}{k\,d_i\,d_j} - 1\right)e^{v_i\,v_j^\top} - \frac{|G|\,\mathbf{A}_{i,j}}{k\,d_i\,d_j} = 0, \tag{21}$$

which has two solutions, $e^{v_i\,v_j^\top} = -1$ and

$$e^{v_i\,v_j^\top} = \frac{|G|\,\mathbf{A}_{i,j}}{k\,d_i\,d_j}, \tag{22}$$

which leads to

$$v_i\,v_j^\top = \log\left(\frac{|G|\,\mathbf{A}_{i,j}}{k\,d_i\,d_j}\right). \tag{23}$$

Accordingly, the GCN model can be unified as the following matrix factorization

$$\mathbf{V}\mathbf{V}^\top = \log\left(|G|\,\mathbf{D}^{-1}\,\mathbf{A}\,\mathbf{D}^{-1}\right) - \log\left(k\right), \tag{24}$$

co-training with the classification loss $l_{\text{class}}$, where node representations in $\mathbf{V}$ are unitized. $\mathbf{D}$ is a diagonal degree matrix with $\mathbf{D}_{i,i} = d_i$. Moreover, according to the conclusion in (Qiu et al., 2018), the unified matrix of GCN in Eq. (24) is as the same as the unified matrix of LINE with the second order proximity (Tang et al., 2015b), except there are two representation matrices in LINE. More specifically, the matrix factorization in Eq. (24) is as the same as LINE with the first order proximity, which is implicit matrix factorization.

Our theoretical analysis on GCN is summarized as:

**Conclusion 2.1** *Given a connected graph $G$ with adjacency matrix $\mathbf{A}$, graph convolutional networks can be unified as implicit matrix factorization with (1) co-training with labeled node classification; (2) unitization of node representations.*

## 3 CUMF ARCHITECTURE

In this section, based on above analysis, we propose an alternative model to GCN named **C**o-training and **U**nitized **M**atrix **F**actorization (CUMF). Figure 2 provides an overview of the proposed CUMF model, which will be described in detail below.

Let $x_i \in \mathbb{R}^d$ denote the feature vector of node $i$ and $f_1$ denote the first MLP in Figure 2. According to our theoretical analysis, given $x_i$, we conduct vector unitization in $f_1(x_i)$ to obtain $v_i$ (the representation of node $i$) as

$$v_i = \frac{f_1(x_i)}{\|f_1(x_i)\|}. \tag{25}$$

According to the theoretical analysis in section 2, the structural part in our proposed CUMF model should be implicit matrix factorization with negative sampling. Thus, $l_{structure}$ can be formulated as

$$l_{structure} = -\sum_{i \in I} \sum_{j \in C_i} \left( \mathbf{A}_{i,j} \log \left( \lambda \left( v_i \, v_j^\top \right) \right) + k \, \mathbb{E}_{j' \sim P_G} \left[ \log \left( \lambda \left( - v_i \, v_{j'}^\top \right) \right) \right] \right). \tag{26}$$

Furthermore, as in GCN, the classification loss $l_{class}$ can be obtained as

$$l_{\text{class}} = \sum_{i \in I_L} CrossEntropy \left( y_i, \text{softmax} \left( f_2(v_i) \right) \right), \tag{27}$$

where $I_L$ is the set of labeled nodes, $y_i$ is the classification label of node $i$ and $f_2$ is the second MLP in Figure 2. Combining Eq. (8), Eq. (26) and Eq. (27), we obtain the loss function of the proposed CUMF model. That is to say, the proposed CUMF model needs to co-train the classification loss $l_{class}$ with the structural loss $l_{structure}$.

In practice, like many semi-supervised models (Weston et al., 2012; Yang et al., 2016), when we do co-training, the structural loss $l_{structure}$ and the classification loss $l_{class}$ are alternately optimized. To be more specific, we frist pick $b$ batches of positive and negative edges, where we sample $k$ additional negative edges for each positive edge according to Eq. (26). For each batch, we take a gradient step for $l_{structure}$. Then we pick a batch of labeled instances and take a gradient step for $l_{class}$. We repeat this process until convergence.

In summary, the proposed CUMF model is based entirely on our theoretical analysis in section 2. Comparing with previous graph modeling methods (Weston et al., 2012; Perozzi et al., 2014; Tang et al., 2015b;a; Grover & Leskovec, 2016; Yang et al., 2016), the unique features of our proposed CUMF method are mainly illustrated in the unitization of node representations, which is derived exactly from our theoretical analysis. Therefore, under the effective guidance of our theoretical analysis on GCN, the proposed CUMF model is extremely clear, concise and reasonable.

## 4 RELATED WORK

GCN (Kipf & Welling, 2017) updates node representations with the aggregation of its neighbors. Based on GCN, Graph Attention Network (GAT) (Veličković et al., 2017) proposes to aggregate neighbors with taking the diversity of nodes into account.

The GCN model needs the whole graph structure. However, in large scale and complex real-world applications, we have millions of nodes and billions of graph edges. Thus, GCN is both time and space consuming, and is hard to perform in constrained memory or distributed computing. Random sampling (Hamilton et al., 2017) and importance sampling (Chen et al., 2018b) propose sampling approaches to reduce the computation of aggregation. Instead of approximating the node representations, variance controlled GCN (Chen et al., 2018a) uses sampled node to estimate the change of node representations in every updating step. Cluster-GCN (Chiang et al., 2019) uses graph partition method to split the whole graph into a set of small sub-graphs, where aggregation happens within each small sub-graph. Thus, Cluster-GCN supports constructing mini-batches and distributed computing.

The original GCN only considers node features, while the features on edges are also important in a graph. To take contextual features on edges into account, RGCN (Schlichtkrull et al., 2018) uses edge types to identify nodes' transition matrix. It achieves good performances in link prediction and entity classification. EGNN (Gong & Cheng, 2019) treats edge features as a tensor and proposes the doubly stochastic normalization to avoid the element values of output features too big. Based on GCN and GAT, EGNN proposes EGNN(C) and EGNN(A) to handle edge tensor respectively.

Table 1: Statistcs of the datasets.

| Dataset | Type | #Nodes | #Edges | #Features | #Classes |
|---|---|---|---|---|---|
| Cora | Citation network | 2,708 | 5,429 | 1,433 | 7 |
| Citeseer | Citation network | 3,327 | 4,732 | 3,703 | 6 |
| Pubmed | Citation network | 19,717 | 44,338 | 500 | 3 |
| BlogCatalog | Social network | 5,196 | 171,743 | 8,189 | 6 |
| Flickr | Social network | 7,575 | 239,738 | 12,047 | 9 |
| USA | Air-traffic network | 1,190 | 13,599 | / | 4 |
| Europe | Air-traffic network | 399 | 5,995 | / | 4 |

## 5  EXPERIMENTS

In this section, we empirically evaluate the performance of the proposed CUMF model in an all-round way. We first describe the datasets and settings of the experiments, then report and analyze the experimental results. Thorough and empirical evaluations are conducted to answer the following research questions:

- **Q1** What are the roles of different components in the CUMF model (co-training & unitization)?

- **Q2** How does the performance of our CUMF model compare to that of GCN on different datasets?

- **Q3** How the two hyper-parameters effect the performance of our method?

- **Q4** Comparing with GCN, is our proposed CUMF model more friendly to distributed computing?

### 5.1  DATASETS

We evaluate our proposed method based on seven datasets. The statistic of the datasets are shown in Table 1. Cora, Citeseer and Pubmed (Sen et al., 2008) are three standard citation network benchmark datasets. BlogCatalog and Flickr (Huang et al., 2017) are social networks. The posted keywords or tags in BlogCatalog and Flickr networks are used as node features. Therefore, these five datasets all have node features. By removing their node features and preserving only the node itself, we made Cora-F, Citeseer-F, Pubmed-F, BlogCatalog-F and Flickr-F. In addition, USA and Europe are two air-traffic networks (Ribeiro et al., 2017), where each node corresponds to an airport and edge indicates the existence of flights between the airports. So these two datasets do not contain node features.

In summary, in terms of whether or not node features are included, we test our model on two types of datasets:

- **Structural Datasets**: Cora-F, Citeseer-F, Pubmed-F, BlogCatalog-F, Flickr-F, USA and Europe.

- **Attributed Datasets**: Cora, Citeseer, Pubmed, BlogCatalog and Flickr.

### 5.2  EXPERIMENTAL SET-UP

Firstly, to verify the roles of co-training and unitization, we include **CUMF-C** (Our method without co-training), **CUMF-U** (Our method without unitization) and **CUMF-C-U** (Our method without co-training and unitization) in our comparisons. Specifically, minus C in name means that the model training is conducted in two stages. That is to say, we first optimize the structure loss independently to obtain the final representations of nodes. Then we keep these representations fixed and only update the parameters of classification model when optimizing the classification loss. Accordingly, minus U in name means that the node representation will not be unitized. Secondly, we compare against the strong baseline (**Planetoid\***) and the state-of-the-art approach (**GCN**) as in (Kipf & Welling, 2017). Lastly, to test the performace of distributed CUMF, we also include the baseline

Table 2: Summary of results in terms of classification accuracy (in percent) on **structural datasets**.

|  | CUMF-C-U | CUMF-C | CUMF-U | CUMF | GCN |
|---|---|---|---|---|---|
| Cora-F | 50.78 | 55.13 | 51.17 | **69.11** | 65.64 |
| Citeseer-F | 33.32 | 35.46 | 35.36 | **46.24** | 44.36 |
| Pubmed-F | 43.62 | 42.23 | 44.41 | **66.15** | 58.26 |
| BlogCatalog-F | 45.76 | 52.94 | 60.85 | **67.64** | 67.07 |
| Flickr-F | 24.61 | 23.03 | 38.62 | **43.16** | 30.86 |
| USA | 42.47 | 50.35 | **60.61** | 58.26 | 53.93 |
| Europe | 41.13 | 40.38 | 50.00 | **50.98** | 41.43 |

Table 3: Summary of results in terms of classification accuracy (in percent) on **attributed datasets**.

|  | Cora | Citeseer | Pubmed | BlogCatalog | Flickr |
|---|---|---|---|---|---|
| Planetoid* | 75.73 | 64.72 | 77.20 | 84.71 | 70.90 |
| GCN | **81.20** | 70.30 | 79.00 | 65.20 | 62.8 |
| CUMF | 81.14 | **70.91** | **80.39** | **91.62** | **77.84** |

(**Random-GCN**) and the state-of-the-art method (**Cluster-GCN**) as in (Chiang et al., 2019) in our comparisons.

We implement our methods on Tensorflow. For the other methods, we use the original papers' code from their github pages. We use the Adam optimizer for all methods with weight decay as zero. And for all methods, we report the mean accuracy of 10 runs with random weight initializations. For our methods: mini-batch size is 256, the dimension of embedding is $0.1*$ the dimension of feature, dropout rate is 0.5, learning rate $\in [0.001, 0.01]$. As mentioned in section 3, there are two other important hyper-parameters: the number of negative edges ($k$) and the balance between structure loss and classification loss ($b$). We will analyze the effects of $k$ and $b$ in next section.

### 5.3 Performance Analysis

Results of our experiments are summarized in Table 2, Table 3, Table 4 and Figure 3, which will be described in detail below.

**The Roles of Co-training and Unitization (Q1)**

Firstly, we mainly examine the roles of different components of our proposed method (co-training & unitization). Specifically, the impact of different components on the capturing of structural information needs to be verified. Thus we conduct experiments on structural datasets. According to Table 2, CUMF for semi-supervised node classification outperforms other versions of CUMF by a significant margin. This fully verifies that unitization and co-training are two essential components of CUMF. Therefore, the correctness of our theoretical analysis is also verified. Meanwhile, to be noted that, CUMF is superior to GCN on all structural datasets. This shows that CUMF is most likely to make better use of structural information than GCN.

**CUMF vs. GCN (Q2)**

After verifying that our proposed CUMF model is not inferior to GCN in capturing structural information, we directly compare CUMF, GCN and Planetoid* on attributed datasets. Based on the results shown in Table 3, the performance of CUMF is consistent to that of GCN on Cora, Citeseer and Pubmed, and it has significant improvement on BlogCatalog and Flickr. It is important to note that the average node degree of BlogCatalog or Flickr is much higher than that of the former three datasets. That is to say, BlogCatalog and Flickr are denser graphs. Therefore, CUMF is consistent with GCN on sparse graphs and significantly outperforms GCN on dense graphs.

Previous work shows that the graph convolution operation is actually a special form of Laplacian smoothing, and starcking many convolution layers may lead to over-smoothing (Li et al., 2018). From this perspective, the reason might be that dense graphs are easier to make GCN becoming over-smoothing than sparse gaphs. As mentioned in section 3, we optimize the structure loss by mini-batch gradient descent. Comparing with the graph convolution operation in GCN, the capturing

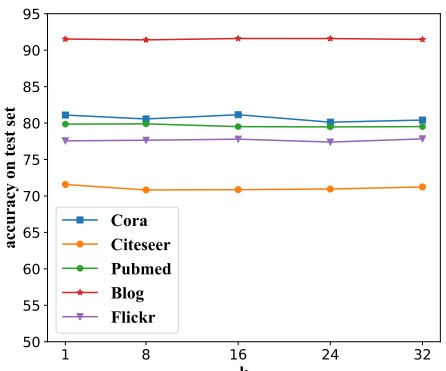 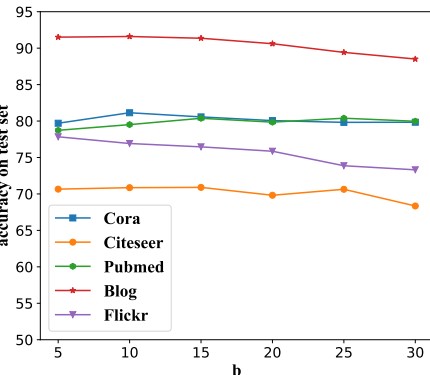

(a) Testing performances w.r.t. the number of negative samples for each positive edge.

(b) Testing performances w.r.t. the balance between structure loss and classification loss.

Figure 3: Testing performances w.r.t parameters introduced by our method (see section 3).

Table 4: Summary of results of distributed methods on **attributed datasets**.

|            | Cora      | Citeseer  | Pubmed    |
|------------|-----------|-----------|-----------|
| Random-GCN | 71.67     | 61.91     | 71.78     |
| Cluster-GCN| 77.10     | 64.87     | 76.60     |
| CUMF       | **81.11** | **70.88** | **80.32** |

of structural information in our approach is much more flexible. Thus, our proposed CUMF model is less likely converge to over-smoothing. The following experimental phenomena in stability analysis also support this view.

**Stability Analysis (Q3)**

We also formally analyze the effects of two important hyper-parameters ($b$ & $k$) on model performance. As mentioned in section 3, $b$ controls the balance between classification loss and structure loss, and $k$ is the number of negative samples for each positive edge. We conduct empirical experiments on attributed datasets, and the results are shown in Figure 3. The flat lines in Figure 3(a) demonstrates that $k$ has little effect on the testing performances of our proposed CUMF model. As shown in Figure 3(b), for sparse datasets, $b$ has little impact on the testing performances of CUMF. For dense datasets, i.e., BlogCatalog and Flickr, the performances of CUMF model decrease with the increasing of $b$. The larger value of $b$ indicates the CUMF model pays more attention on capturing structural information. This may give a reason of the poor performances of GCN on dense datasets: the graph convolution operation is easily to be extremely over-smoothing for capturing structural information on dense datasets. On the other hand, CUMF is flexible with the sparsity of graphs via adjusting the parameter $b$, which is relatively stable.

**Distributed CUMF vs. Distributed GCN (Q4)**

In the end, we verify the capacity of CUMF (our proposed method) and GCN on distributed computing in our experiments. Specifically, as the performances of centralized GCN is close to those of centralized CUMF on Cora, Citeseer and Pubmed, we only conduct empirical experiments on these datasets for convenience of comparisons. And we include Random-GCN and Cluster-GCN as baselines. Besides, our distributed experiments are conducted in synchronous mode. Experimental results are shown in Table 4. On these datasets, distributed CUMF does not suffer performance loss, but distributed GCN methods greatly do, comparing with results in Table 3. The reason is that CUMF is MF-based, and MF-based methods naturally support constructing mini-batches (Zhuang et al., 2013; Yu et al., 2014). However, for GCN, constructing mini-batches is equivalent to solving graph partitioning problem (Chiang et al., 2019), which is actually a quite challenging work

(Karypis & Kumar, 1998; Dhillon et al., 2007). Moreover, it is most likely that only a small portion of each mini-batch is labeled, which may greatly affect the convergence of GCN.

## 6 CONCLUSION

To the best of our knowledge, CUMF is the first work that connects GCN to MF. We theoretically unify GCN as co-training and unitized matrix factorization, and a CUMF model is therefore proposed. We conduct thorough and empirical experiments, which strongly verify the correctness of our theoretical analysis. The experimental results show that CUMF achieve similar or superior performances compared to GCN. We also observe that GCN performs poor on dense graphs, while CUMF has great performances. This is may caused by the over-smoothing of graph convolution on dense graphs, while CUMF can balance the smoothing of neighbours and the classification of labeled nodes via co-training. Moreover, due to the MF-based architecture, CUMF is extremely flexible and easy to be applied to distributed computing for large scale real-world applications, and significantly outperforms state-of-the-art distributed GCN methods.

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
