# OpenReview forum: "Unifying Graph Convolutional Networks as Matrix Factorization"
_ICLR.cc/2020/Conference — Reject_

### Official Review · AnonReviewer3 · 2019-10-11
**Official Blind Review #3**

**Rating:** 6

**Review:**

In this paper, the authors propose a new method for semi-supervised node classification by drawing connection between GCN and MF. The authors borrow the idea of convergence of GCN as Laplacian Smoothing. With this observation, the authors propose a joint loss with two components: classification loss and structure loss for the similarity between embedding of neighboring nodes. The authors train the parameters via optimizing the two losses alternatively. Experiments are carried out on seven networks with comparison to baselines.

Strength:
1. It is an interesting and innovative idea to draw connection between GCN and MF.
2. The propose method is more suitable for distributed setting. With negative sampling for structure loss, both structure batch and classification batch can be constructed locally with only one-hop information.
3. The authors carry out experiments on seven real-world networks with ablation study for components in the model. Moreover, the authors carry out comparison to baselines in distributed setting.

Weakness:
1. The connection of GCN to MF is very indirect. It holds only when the GCN converges to the Laplacian smoothing. It is not clear whether this holds empirically. Moreover, there are too much intermediate steps and approximation between the Laplacian smoothing to the matrix factorization. As far as I am concerned, the connection is closer to node embedding versus matrix factorization.
2. Given that GCN serves as Laplacian smoothing, it would be great if the authors can simply add additional regularization on dis(h_i, h_i) for (v_i, v_j)\in E. Moreover, there is no reference and description to the Planetoid* algorithm.
3. The authors use alternative batches between structure and classification loss. It would be interesting to see if joint training the two loss in mini-batch among a node and its neighbors can leads to any difference.
3. The authors report only accuracy as evaluation metrics. It would be better If the authors could report recall@K and F1 score as well.


**Experience Assessment:**

I have published one or two papers in this area.

**Review Assessment: Checking Correctness Of Derivations And Theory:**

I carefully checked the derivations and theory.

**Review Assessment: Checking Correctness Of Experiments:**

I carefully checked the experiments.

**Review Assessment: Thoroughness In Paper Reading:**

I read the paper thoroughly.

---

> ### Author Response · Authors · 2019-11-11
> **Thanks for your comments.**
>
> Thanks for your comments. Here are our replies:
>
> (1) Yes, the connection is not quiet direct. The aim of our work is to give a MF-based form to GCN for flexible modeling in real-world large-scale applications. Finding an extreme equivelant alternative model of GCN is not our real purpose. And node embedding methods can be unified as MF in previous works [1], so GCN is related to node embedding as well as MF.
>
> (2) Yes, this will be futher investigated in our future works.
>
> (3) Actually, the two methods achieve similar accuracy. And the method with alternative batches between structure and classification loss can converge easier. The difference is not significant.
>
> (4) As only accuracy is reported in previous works, so we followed this setting in our work.
>
>
> [1] Qiu J, Dong Y, Ma H, et al. Network embedding as matrix factorization: Unifying deepwalk, line, pte, and node2vec[C]//Proceedings of the Eleventh ACM International Conference on Web Search and Data Mining. ACM, 2018: 459-467.

---

> > ### Comment · AnonReviewer3 · 2019-11-12
> > **Thank you for the response.**
> >
> > Thank you for the response. However, the arguments about the connection to MF is not very strong. I will remain my initial evaluation.

---

### Official Review · AnonReviewer1 · 2019-10-22
**Official Blind Review #1**

**Rating:** 1

**Review:**


The work poses an interesting question: Are GCNs (and GNNs) just special types of matrix factorization methods? Unfortunately, the short answer is **no**, which goes against what the authors say.

Until recently I thought like the authors, but the concurrent work [1] (On the Equivalence between Node Embeddings and Structural Graph Representations) https://openreview.net/forum?id=SJxzFySKwH changed my mind.
The work of (Li et al., 2018) shows that nearby nodes tend to get similar representations. There is mounting experimental evidence of that being the case in real-world graphs (e.g., https://arxiv.org/abs/1908.08572). But [1] shows that GCNs and GNNs are fundamentally different from matrix factorization methods, regardless of the loss function used to learn the embeddings. Consider Figure 1 in [1], and it is easy to see that matrix factorization will give different embeddings to the Lynx and Orca nodes, while GCNs and GNNs must give the same embedding. Even if we connect the graphs through the Spruce and the Zooplankton nodes, their conclusion would not change. Matrix factorization (as broadly understood) will give embeddings that can even be used to cluster nodes. The eigenvectors of the symmetric Laplacian encode the diffusion of a type of random walk and nodes that are far away in the graph must have different embeddings (because through the diffusion operator, they are far away).

In GCNs, the convergence of the embeddings is better explained by the mixing of a random walk (Theorem 1 of (Xu et al., 2018)), which, in the special case of a GCN, converges to 1/sqrt(degree of node), as shown by (Li et al., 2018) in their Theorem 1 for the symmetric Laplacian. This is unrelated to what we get in matrix factorization as explained earlier.

What is wrong with the math: Equation (11) is equated with matrix factorization, but note that it does not account for nonedges, while matrix factorization accounts for nonedges. This issue is more clear in Equation (14). The problem happens when the paper jumps from Equation (14), which is correct but not MF, to Equation (15) which is MF but unrelated to Equation (14). The argument is that “negative edges sampling is used, for better convergence”… sorry, not for better convergence, it completely changes the optimization objective. Hence, GCNs are not matrix factorization methods.

I think the paper is a valiant effort, but unfortunately the core premise is incorrect. The jump from Equation (14) to equation (15) cannot be justified, and I believe showcases a fundamental flaw the argument. I do not see a way to fix the paper. I vote to reject it.

[1] On the Equivalence between Node Embeddings and Structural Graph Representations, https://openreview.net/forum?id=SJxzFySKwH
Xu, K., Li, C., Tian, Y., Sonobe, T., Kawarabayashi, K.I. and Jegelka, S., 2018. Representation learning on graphs with jumping knowledge networks. ICML 2018.
Li, Qimai, Zhichao Han, and Xiao-Ming Wu. Deeper insights into graph convolutional networks for semi-supervised learning. AAAI, 2018.


--------------

Read rebuttal. Will keep my original assessment.


**Experience Assessment:**

I have published in this field for several years.

**Review Assessment: Checking Correctness Of Derivations And Theory:**

I carefully checked the derivations and theory.

**Review Assessment: Checking Correctness Of Experiments:**

I assessed the sensibility of the experiments.

**Review Assessment: Thoroughness In Paper Reading:**

I read the paper thoroughly.

---

> ### Author Response · Authors · 2019-11-11
> **Thanks for your comments.**
>
> Thanks for your comments.
>
> I agree that the conclusion in our work is not an extreme equivelant form of GCN, but I don't think it is "incorrect". The aim of our work is to give a MF-based form to GCN for flexible modeling in real-world large-scale applications. Finding an extreme equivelant alternative model of GCN is not our real purpose. And according to our observation, this purpose is satisfied. According to our experimental results, MF is good enough for node classification in all kinds of settings, and we don't need to use GCN in real-world applications. I think this is a valuable conclusion of our work.

---

### Official Review · AnonReviewer2 · 2019-10-22
**Official Blind Review #2**

**Rating:** 1

**Review:**

This paper derives a matrix-factorization approach for node classification. The approach is closely related to GCN. The authors show that the proposed approach outperforms GCN and Planetoid empirically.

Though empirically appealing, this paper has a few pitfalls that need be addressed.

1. The wording "unifying" is a misnomer. The title "unifying graph convolutional networks" hallucinates a framework that unifies several neural network architectures, which is not precise. In reality, the authors propose a learning objective that consists of two loss terms, the classification loss and the structure loss. The classification loss is nothing but the usual GCN. The structure loss is the contribution of the paper. The derivation of this term starts from GCN and a Laplacian smoothing argument, and arrives at a matrix factorization form through a series of modeling modifications. By and large, the title is misleading.

2. The wording "correctness of our theoretical analysis" is dubious. The paper does not present a theoretical analysis. The derivation of the matrix factorization is only a modeling process. In no mathematical sense the factorization is equivalent to GCN.

3. The alternating training is questionable. The authors propose alternately optimizing the structure loss and the classification loss. Since taking the gradient of the whole loss function is straightforward in all graph neural network approaches, it is unclear why the authors prefer the alternating optimization approach. Supplementing a convergence plot and comparing the two approaches may help, if the alternating approach is indeed better.

4. The "distributed computing" component needs more substantiation. It is unclear whether this phrase actually means the concept familiar by the parallel computing community. Therein, computation is done by using several machines communicated through networked protocols. Machine setting, parallel implementation details, and speedup are the primary interests in distributed computing. All information should be reported.

Questions:

1. First sentence of section 5. What does "all-round" mean?

2. Stability Analysis. What is b? The reader does not find a definition elsewhere. A probably related concept is alpha (see eqn (8)).

3. Figure 3(b) shows that larger b leads to poorer performance. The authors state that a larger b means a stronger emphasis on the structure loss. Consequently, it appears that putting more emphasis on the structure term leads to poorer performance. Then, does it mean that the structure term is a useless contribution?


**Experience Assessment:**

I have published one or two papers in this area.

**Review Assessment: Checking Correctness Of Derivations And Theory:**

I carefully checked the derivations and theory.

**Review Assessment: Checking Correctness Of Experiments:**

I carefully checked the experiments.

**Review Assessment: Thoroughness In Paper Reading:**

I read the paper at least twice and used my best judgement in assessing the paper.

---

> ### Author Response · Authors · 2019-11-11
> **Thanks for your comments.**
>
> Thanks for your comments.
>
> [1-2] We are sorry that our expression might be misleading. And I agree that the conclusion in our work is not an extreme equivelant form of GCN. The aim of our work is to give a MF-based form to GCN for flexible modeling in real-world large-scale applications. Finding an extreme equivelant alternative model of GCN is not our real purpose. And according to our observation, this purpose is satisfied. According to our experimental results, MF is good enough for node classification in all kinds of settings, and we don't need to use GCN in real-world applications. I think this is a valuable conclusion of our work.
>
> (3) This is incorporated following previos work [1].
>
> (4) We ran the models on three workers. The speedup is not our focus in this work. We only want to show that batched or distributed GCN suffers from performance loss.
>
> [1] Weston J, Ratle F, Mobahi H, et al. Deep learning via semi-supervised embedding[M]//Neural Networks: Tricks of the Trade. Springer, Berlin, Heidelberg, 2012: 639-655.

---

> > ### Comment · AnonReviewer2 · 2019-11-11
> > **Thank you for the response.**
> >
> > Thank you for the response. While I appreciate the empirical results, the arguments however are not strong enough to override the original assessment. Hope the work can be improved addressing the four critical points mentioned in the review.

---

### Public Comment · ~Xiao_Qi2 · 2019-10-11
**Interesting, but I have something to discuss.**

Great work. If the claims in this paper can work, the most exciting thing is, we can easily perform semi-supervised learning on large graphs, instead of using GCNs. Here, I have several questions:

GCNs usually have two different forms:
$\mathop {\rm{H}}\nolimits^{l + 1}  = \sigma \left( {\mathop {\tilde D}\nolimits^{ - 1/2} \tilde A\mathop {\tilde D}\nolimits^{ - 1/2} \mathop H\nolimits^l \mathop W\nolimits^l } \right)$
and
$\mathop {\rm{H}}\nolimits^{l + 1}  = \sigma \left( {\mathop {\tilde D}\nolimits^{ - 1} \tilde A\mathop H\nolimits^l \mathop W\nolimits^l } \right)$
Will the two forms affect the unification?

Can Euclidean distance have similar performance as in Fig. 1?

Considering GCNs can be unified as MF, and this achieves better performances, can other graph embedding methods (like LINE and DeepWalk) do this? And why there are no previous work using these graph embedding methods to beat GCNs?

---

> ### Author Response · Authors · 2019-10-12
> **Thanks for your reply.**
>
> Thanks for your reply.
>
> Here are our responses:
>
> (1)	During the unification, the difference between the two forms of GCN only lies in different $\beta_{i,j}$ in Eq. (14). And the $\beta_{i,j}$ can be eliminate in the following deduction, and we can achieve the same unified matrix as in Eq. (24). Thus, the two forms of GCN does not affect our unification.
>
> (2)	We have tried the Euclidean distance. And the Euclidean distance is totally not consistent with the convergence of GCN.
>
> (3)	Yes, other graph embedding methods can do this.
> The reason of “why there are no previous work using these graph embedding methods to beat GCNs” is, there lacks co-training and unitizing process in existing graph embedding methods. According to our theoretical analysis, we add co-training and unitization in MF, and obtain our CUMF model. According to the results in table 2, CUMF without co-training or unitization (i.e., CUMF-C or CUMF-U) is totally not competitive comparing with CUMF. Thus, co-training or unitization are extremely important for graph embedding methods to achieve good performances in semi-supervised learning.
> Moreover, Planetoid can be somehow viewed as a co-training DeepWalk, without the process of unitization. It is the most competitive baseline for GCN, and even beats GCN on the BlogCatalog dataset and the Flickr dataset according to the results in table 3.

---

> > ### Public Comment · ~Xiao_Qi2 · 2019-10-12
> > **More discussion**
> >
> > Thanks.
> >
> > So, with cotraining and unitizing, other graph embedding methods, like LINE, DeepWalk and Node2Vec, can also perform well on semi-supervised learning?

---

> > > ### Author Response · Authors · 2019-10-12
> > > **Yes**
> > >
> > > Yes, according to our theoretical analysis, and analysis between graph embedding and MF, this is of great possibility.

---

### Decision · Program_Chairs · 2019-12-19

**Decision:**

Reject

**Comment:**

The paper makes an interesting attempt at connecting graph convolutional neural networks (GCN) with matrix factorization (MF) and then develops a MF solution that achieves similar prediction performance as GCN.

While the work is a good attempt, the work suffers from two major issues: (1)  the connection between GCN and other related models have been examined recently. The paper did not provide additional insights; (2) some parts of the derivations could be problematic.

The paper could be a good publication in the future if the motivation of the work can be repositioned.